# Early-stage lung cancer is driven by a transitional cell state dependent on a KRAS-ITGA3-SRC axis

Aaron L Moye[1,2,3], Antonella FM Dost[1,2,3,4], Robert Ietswaart [3], Shreoshi Sengupta[1,2,3], VanNashlee Ya[1,2,3], Chrystal Aluya [1,2,3], Caroline G Fahey [1,2,3,5], Sharon M Louie[1,2,3], Margherita Paschini[1,2,3] & Carla F Kim [1,2,3 ✉]

## Abstract

Glycine-12 mutations in the GTPase KRAS (KRAS[G12]) are an initiating event for development of lung adenocarcinoma (LUAD). KRAS[G12] mutations promote cell-intrinsic rewiring of alveolar type-II progenitor (AT2) cells, but to what extent such changes interplay with lung homeostasis and cell fate pathways is unclear. Here, we generated single-cell RNA-seq (scRNA-seq) profiles from AT2-mesenchyme organoid co-cultures, mice, and stage-IA LUAD patients, identifying conserved regulators of AT2 transcriptional dynamics and defining the impact of KRAS[G12D] mutation with temporal resolution. In AT2[WT] organoids, we found a transient injury/plasticity state preceding AT2 self-renewal and AT1 differentiation. Early-stage AT2[KRAS] cells exhibited perturbed gene expression dynamics, most notably retention of the injury/plasticity state. The injury state in AT2[KRAS] cells of patients, mice, and organoids was distinguishable from AT2[WT] states via altered receptor expression, including co-expression of ITGA3 and SRC. The combination of clinically relevant KRAS[G12D] and SRC inhibitors impaired AT2[KRAS] organoid growth. Together, our data show that an injury/plasticity state essential for lung repair is co-opted during AT2 self-renewal and LUAD initiation, suggesting that early-stage LUAD may be susceptible to interventions that target specifically the oncogenic nature of this cell state.

**Keywords** Lung; Adenocarcinoma; AT2; KRAS; Cell States
**Subject Categories** Cancer; Molecular Biology of Disease; Respiratory System

## Introduction

LUAD is the leading cause of cancer-associated death worldwide (Sung et al, 2021). Oncogenic mutations in KRAS are present in 30% of LUAD patients (Cancer Genome Atlas Research Network, 2014; Cook et al, 2021). KRAS[G12] variants are sufficient to initiate LUAD in genetically engineered mouse models (GEMMs) and accurately model human LUAD (Jackson et al, 2001). Surfactant-producing AT2 cells have important progenitor functions during injury repair via differentiation into alveolar type I (AT1) cells, the pneumocytes that perform gaseous exchange. Recent studies have defined injury-associated intermediate states in AT2-to-AT1 differentiation (Choi et al, 2020; Kobayashi et al, 2020; Strunz et al, 2020). We previously showed that KRAS[G12D] causes transcriptome rewiring in AT2 cells during LUAD development (Dost et al, 2020). Others have also examined the transcriptional (Wang et al, 2021; Marjanovic et al, 2020), epigenetic (LaFave et al, 2020), and metabolic (Nie et al, 2021) landscapes of different stages of LUAD. However, there are still significant questions regarding early-stage LUAD, such as the interplay between pathways used by AT2 cells in homeostasis, repair, and oncogenesis and how cell-intrinsic KRAS-driven changes alter signaling with the microenvironment is largely unknown.

Identifying cellular states at different stages of LUAD using powerful model systems may provide insight into the mechanisms of LUAD progression and novel targets for therapeutic intervention. Small molecule inhibitors directly targeting oncogenic KRAS[G12C], such as adagrasib and sotorasib, MRTX113 targeting KRAS[G12D], and numerous other pan or variant-specific KRAS-targeting drugs are entering the clinic (Punekar et al, 2022; Kim et al, 2023). The discovery of variant-specific KRAS inhibitors is a major breakthrough, yet clinical trial data has revealed that only a subset of indicated KRAS mutant patients respond to treatment (Fakih et al, 2022; Skoulidis et al, 2021), highlighting the need to find novel combination treatments to improve efficacy and overcome resistance mechanisms. Significant improvements in early-stage cancer detection using low-dose computed tomography (LDCT) (Jonas et al, 2021) and blood-based detection assays (Cohen et al, 2018; Lennon et al, 2020) will ultimately result in more patients being diagnosed, increasing the demand for early-stage therapeutics. To address these gaps, we sought to identify transcriptional dynamics downstream of activation of oncogenic KRAS expression in AT2 cells.

[1]Stem Cell Program and Divisions of Hematology/Oncology and Pulmonary Medicine, Boston Children's Hospital, Boston, MA, USA. [2]Harvard Stem Cell Institute, Cambridge, MA, USA. [3]Department of Genetics, Harvard Medical School, Boston, MA, USA. [4]Present address: Hubrecht Institute, Oncode Institute, Royal Netherlands Academy of Arts and Sciences (KNAW), Utrecht, The Netherlands. [5]Present address: Harvard University and Department of Medical Oncology, Dana-Farber Cancer Institute, Boston, MA, USA. ✉E-mail: carla.kim@childrens.harvard.edu

# Results

## Lung AT2 progenitors activate a transient injury/plasticity state prior to both AT2 self-renewal and AT1 differentiation

To identify transcriptional changes specific for KRAS-mutant LUAD initiation at the earliest stages, we generated and analyzed scRNA-seq data from control $Rosa26^{YFP}$ AT2 organoids (henceforth, AT2$^{WT}$), $Kras^{G12D/WT}$ $p53^{flox/flox}$ $Rosa26^{YFP}$ (henceforth, AT2$^{KRAS}$) AT2 organoids, and their co-cultured non-cancerous mesenchymal cells. We used our previously described 3D organoid culture combined with Cre-mediated, adenoviral in vitro induction (Fig. 1A) (Dost et al, 2020). AT2 (YFP$^{POS}$) organoids and mesenchymal (YFP$^{NEG}$ EPCAM$^{NEG}$) cells were collected using fluorescence-activated cell sorting (FACS) at 4, 7, 10, and 14 days after infecting cells with Ad5-CMV-Cre (Fig. EV1A). The scRNA-seq data was generated using the 10× Genomics 3′ platform and preprocessed for further analysis ($n = 71{,}252$) (Fig. EV1B–D). Transcriptionally distinct states were identified using differential expression (DE) analysis on the combined AT2$^{WT}$ and AT2$^{KRAS}$ organoid data binned by genotype and time point (Fig. EV1D and Dataset EV1). After DE genes were filtered for transcription factors (TFs) we observed that cells from each time point had a distinct TF profile, confirming that both AT2$^{WT}$ and AT2$^{KRAS}$ organoids are transcriptionally rewired in a temporally-defined manner (Fig. EV1E). We previously showed that $SOX9$ expression and loss of AT2 marker gene expression are hallmark transcriptional changes in AT2 cells early after $Kras$ mutation (Dost et al, 2020). In agreement, $Sox9$ expression was absent in all AT2$^{WT}$ organoid time points, was expressed exclusively in AT2$^{KRAS}$ organoids after Day 7, and correlated with a decreased expression of AT2 lineage-defining genes $Etv5$, $Lyz2$, and $Nkx2.1$ (Fig. 1B). We benchmarked our time course data by comparing the expression of genes reported to promote tumorigenesis in $Kras$-mutant GEMMs. In AT2$^{KRAS}$ organoids, $Tigit$, $Cldn4$, and $Itga2$, markers of a high plasticity cell state (HPCS) (Marjanovic et al, 2020), and $Lgr5$ were expressed at all AT2$^{KRAS}$ time points (Fig. 1B) (Tammela et al, 2017). AT2$^{KRAS}$ organoids also expressed $Krt8$, a marker of the transitional injury-associated cell state between AT2 cells and AT1 cells, observed during lung regeneration (Strunz et al, 2020). We noticed that many of the HPCS genes are also genes that comprise the injury-associated transitional state and thus termed this an injury/plasticity state. Examining this set of genes, we noted their transient expression in AT2$^{WT}$ organoids, particularly present at Day 4 and absent at later time points when cells with high expression of either AT2 or AT1 cell signature genes were present. In contrast, AT2$^{KRAS}$ organoid cells retained this injury/plasticity state across all time points (Fig. 1B).

To further examine the transcriptional changes of AT2$^{WT}$ cell states, we subset the epithelial component of our AT2$^{WT}$ organoid data and performed Leiden community detection (Figs. 1C and EV2A). AT2$^{WT}$ organoids initially formed a transcriptionally homogenous cell state at Day 4 before establishing two distinct terminal states by Day 14. Over the 14 days examined, two distinct AT2$^{WT}$ endpoints formed with either an AT1 cell (Leiden community C6) or AT2 cell signature (C1) with distinct intermediate cell states (C5 and C7) preceding both fates, respectively (Figs. 1D–F and EV2B,C). Along the AT1 fate (C5

and C6), we observed the expression of $Hopx^{POS}$ $Igfbp2^{NEG}$ and $Hopx^{POS}$ $Igfbp2^{POS}$ cells, which are markers for stem-like and terminally differentiated AT1 cells, respectively (Wang et al, 2018), and cells within the AT2 fate had little to no expression of either gene (Fig. 1E). We found elevated $Tead1$ and $Wwrt1$ (encodes for TAZ) expression in the AT1 fate, consistent with recent observations that YAP/TAZ has a vital role in AT1 cell identity (Penkala et al, 2021; Burgess et al, 2024) (Fig. EV2D). Furthermore, the early-stage Day 4 AT2$^{WT}$ state (C3) had little to no AT1 or AT2 signature score indicating that AT2$^{WT}$ cells enter a lineage plastic state as part of homeostatic regeneration (Fig. 1F).

Next, we further investigated the potential relationships between AT2$^{WT}$ organoid states. First, we examined the association between our temporally defined states and recently discovered AT2 injury states, identified as precursors to AT1 differentiation (Choi et al, 2020; Strunz et al, 2020). Our analysis revealed that "DATP" ($Cldn4$, $Krt8$, $Ndgr1$, $Sprr1a$), "IFNγ" ($Ifgnr1$, $Ly6a$, $Irf7$, $Cxcl16$), and "Primed AT2" ($Nr4a3$, $Tmem173$, $Orm1$, $Cbr4$) gene signatures were activated in early (C3) and intermediate (C5 and C7) AT2$^{WT}$ states but were suppressed by Day 14 (Fig. 1G). Furthermore, these injury signatures negatively correlated with AT2 signature expression, suggesting that injury signatures are a precursor for AT2 self-renewal and correlates with lineage plasticity. We next tested the fate commitment probabilities of early (C3) and intermediate (C5 and C7) states in AT2$^{WT}$ organoids. We computationally determined the transition probability of C5 and C7 using Palantir, which captures the continuity in cell states modeled as a stochastic process, then assigns a probability of differentiating into predefined terminal states (Setty et al, 2019). Start and end points were defined using our temporal information and gene expression, with C3 (Day 4) being the start point and C6 (AT1, Day 14) and C1 (AT2, Day 14) being the endpoints. C3 had approximately a 50% probability of transitioning to the AT1 or AT2 state, supporting our model that a lineage-plastic state is found in AT2$^{WT}$ organoids with AT1 and AT2 differentiation potential. For the intermediate states, the majority of the C7 state were predicted to transition to the AT2 state; in contrast, nearly all the C5 state were predicted to transition to the AT1 state (Fig. 1H). Hence, within our temporal AT2$^{WT}$ organoid data we identified early-stage activation of a transient, injury/plasticity (AT2-sig$^{LOW}$ AT1-sig$^{LOW}$) state which precedes the formation of fate-committed intermediate states and endpoint AT2 and AT1 states. Lastly, we sought to better understand the TF regulators of fate choice in AT2$^{WT}$ organoids. We subset AT1 and AT2 intermediate states, performed DE analysis, and filtered for TFs ($p$ value < 0.05 and log$_2$ fold change >1) using a custom-curated reference list (Hu et al, 2019; Schmeier et al, 2017) (Dataset EV2). $Nfe213$ was amongst the top DE TFs in C5 and was previously identified as DE in E15–E18 pre-ATI cells (Gonzalez et al, 2019), consistent with C5 activating a $Hopx^{POS}$ $Igfbp2^{NEG}$ AT1 state. In contrast, the AT2-lineage defining TF $Etv5$ was DE in C7 (Zhang et al, 2017) (Fig. EV2F). Collectively, our temporal data supports an unappreciated model where AT2$^{WT}$ cells activate a transient, injury/plasticity state defined by AT2-lineage suppression and activation of injury-responsive gene programs, followed by the commitment to an AT2 or AT1 fate outcome via transcriptionally distinct AT2- and AT1-intermediate states. We also observed no evidence of Day 14 AT2 organoids transdifferentiating to the AT1 state based on the FA2 representation of the data, arguing that the injury/plasticity state is an important

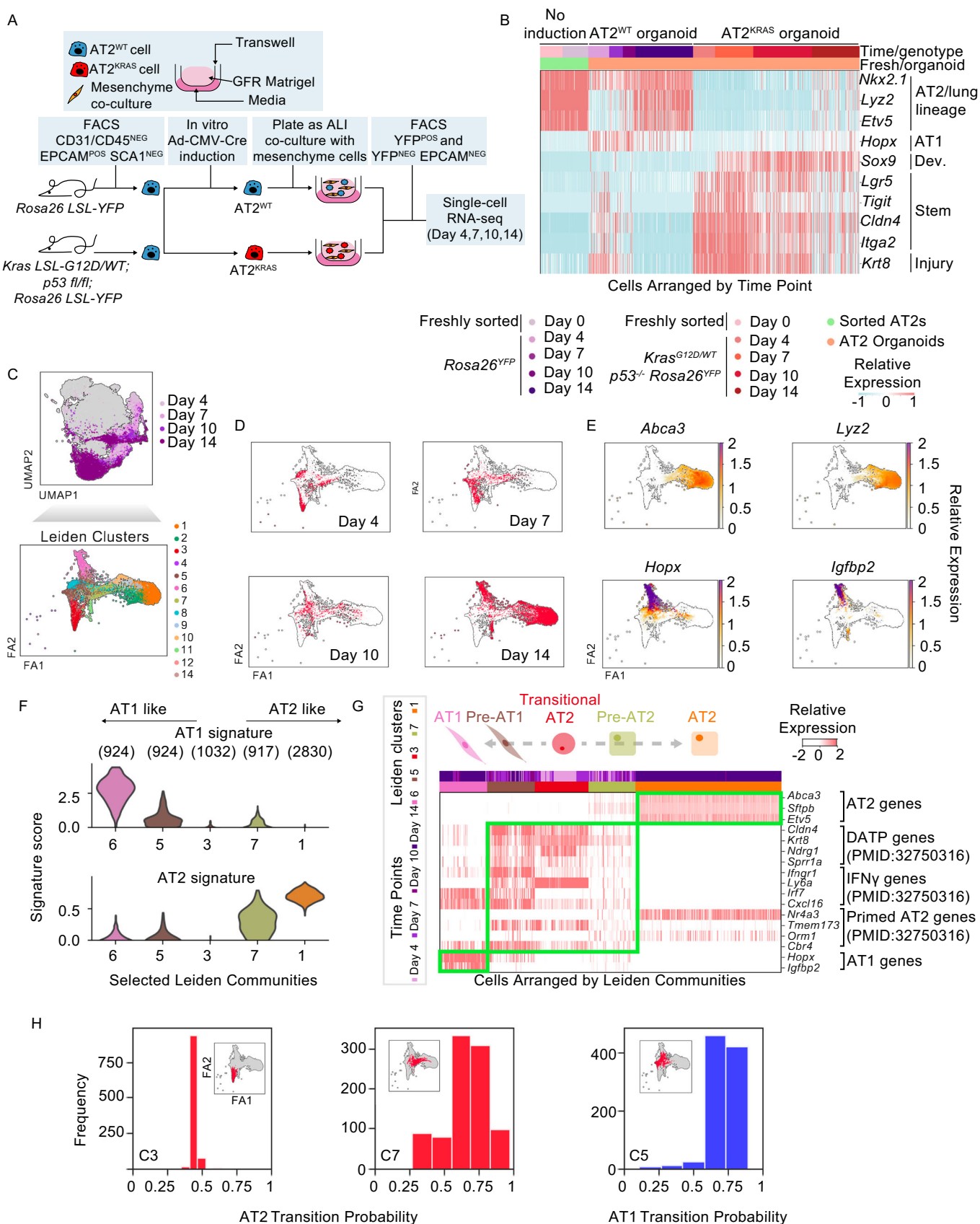

**Figure 1.  Lung AT2 progenitors activate a transient injury/plasticity state prior to both AT2 self-renewal and AT1 differentiation.**

(A) Schematic representation of the AT2-mesenchyme organoid co-culture time course experiment. (B) Heatmap of gene expression in organoid time course data relevant to AT2, AT1, development, stem cell, and injury response states. "No induction" are primary AT2 cells collected from each mouse genotype (Y and KPY) prior to viral induction or culturing as organoids. (C) FA2 representation of filtered single cells subset from AT2^WT organoid data and their corresponding Leiden community. The FA2 in Fig. 1C is reused in Figs. 1D, E, 4A, EV1D, EV2D, and EV5A to show different aspects of the data. (D) FA2 representation of filtered single cells subset from AT2^WT organoid data and their corresponding time point. (E) FA2 representations of filtered single cells subset from AT2^WT organoid data and the relative expression of either AT1 or AT2 specific genes. (F) AT2 and AT1 signature score level per community using a violin plot (y-axis, signature score; x-axis, Leiden community). (n) denotes the number of cells per cluster. (G) Heatmap of relative gene expression in the AT2^WT organoid time course data relevant to AT2, AT1, and injury response gene expression signatures. Cells are arranged with C3 (Day 4 AT2^WT organoids) in the middle, left is the AT1 fate path, and right is the AT2 fate path. (H) Transition probabilities of cells in selected AT2^WT Leiden communities (x-axis; labeled transition probability, y-axis; binned frequency of occurrence). Inset highlights the cell states whose probabilities are being represented using a FA2 representation.

intermediate step in AT2^WT self-renewal and AT1 differentiation (Fig. 1E).

## Kras^G12D promotes distinct temporal subpopulations defined by AT2-lineage and injury/plasticity signatures

Having identified an injury/plasticity state in AT2^WT cells undergoing self-renewal, we wanted to determine if this state is co-opted during tumorigenesis. We analyzed the AT2^KRAS subset of the organoid data, re-processed, and identified Leiden communities (Figs. 2A and EV3A and Dataset EV3). At Day 4, AT2^KRAS organoid cells formed a single community (C4) similar to AT2^WT organoids. However, unlike AT2^WT cells, AT2^KRAS cells had dysregulation in subsequent time points, lacking AT2 or AT1 fate commitment (Fig. 2B). Notably, AT1 marker genes *Hopx* and *Igfbp2* as well an AT1 signature were suppressed in AT2^KRAS cells, and AT2 marker genes (*Abca3*, *Lyz2* and a multiple-gene AT2 signature) were only robustly expressed in a subpopulation of Day 14 organoid cells (Fig. 2B–D). In contrast to AT2^WT cells with transient expression of injury/plasticity genes during organoid culture, these transcriptional signatures were prolonged beyond Day 7 in AT2^KRAS organoids (Fig. 2E). At Day 14, AT2^HIGH C5 cells re-activated canonical AT2 genes *Abca3*, *Sftpb*, and *Etv5* while suppressing injury/plasticity genes (Fig. 2E). In contrast, Day 14 AT2^LOW C2 cells maintained the lineage plastic state characterized by continued expression of injury response genes, suppression of AT2 genes, and elevated expression of *Sox9* (Figs. 2E and EV3B). To further examine if injury/plasticity and AT2 lineage signature can be used to temporally define AT2^KRAS heterogeneity, we subset mid- and late-stage AT2^KRAS populations and calculated signature scores. Indeed, distinct cell states were identified in AT2^KRAS progression based on injury/plasticity and AT2 signature. Furthermore, cells with high injury/plasticity signature had low AT2 signature (Fig. 2F). Lastly, we determined the probabilities of AT2^KRAS cell states to form either the AT2 low or high states, where the Day 4 C4 state was defined as the starting point. The vast majority of Day 4–10 states, both injury/plasticity^HIGH and AT2^HIGH, were predicted to transition to Day 14 AT2^LOW state, suggesting that reactivation of the AT2^HIGH state is not an intrinsically favorable outcome in AT2^KRAS progression (Fig. 2G); Despite this, reactivation of the AT2^HIGH state was a strong feature in the latest time points in our study, suggesting non-cell-intrinsic influence on cell states. Overall, KRAS-mutant AT2 cells co-opt the injury/plasticity state beyond the transient nature observed in AT2^WT organoids, resulting in distinct cellular states across early-stage and late-stage time points, delineated by injury/plasticity or AT2-lineage signatures.

## AT2^KRAS states are temporally conserved in vivo and in stage IA human LUAD

Having identified an injury/plasticity state in AT2^WT self-renewal and in the context of oncogenic *Kras* in orgnoids, we sought to validate our observation in vivo by generating a scRNA-seq time course dataset using the *Kras^G12D/WT p53^flox/flox Rosa26^YFP* GEMM. Mice were treated intratracheally with an Ad5-SPC-Cre, resulting in tumor initiation specifically in AT2 progenitors (Fig. 3A). AT2 cells were collected using FACS at 4, 7, 10, and 16 weeks post-infection, covering the transition from hyperplasia to adenocarcinoma (Jackson et al, 2001) (Fig. EV4A). Furthermore, mice were induced so that AT2 cells could be collected and scRNA-seq libraries generated at the same time (materials and methods), allowing for direct comparison of time points without the need for batch correction. The scRNA-seq libraries were generated using the 10× Genomics platform, and the data processed for downstream analysis (n = 3125 cells) (Figs. 3B and EV4B). Consistent with our observations in organoids, AT2 lineage gene expression negatively correlated with development and injury response genes (Fig. EV4C). To further examine temporal AT2^KRAS heterogeneity, Leiden communities were identified (Figs. 3B and EV4D). Consistent with our organoid data, at the late-stage week 16 time point we observed the formation of distinct AT2^HIGH (C7) and AT2^LOW (C3) states (Fig. 3C). AT2^LOW cells had elevated injury-associated gene expression while AT2^HIGH cells suppressed injury-associated gene expression (Fig. 3C). Next, we wanted to determine if injury/plasticity and AT2 signatures can define distinct mid- and late-stage cell states in vivo. In agreement with our organoid data, injury/plasticity^HIGH AT2^LOW and injury/plasticity^LOW AT2^HIGH populations were observed at mid and late-stage time points (Fig. 3D) confirming a conserved interplay between injury/plasticity and AT2-lineage gene expression dynamics in LUAD progression. Lastly, to establish human relevance we reanalyzed our scRNA-seq data from two stage IA, Kras-mutant LUAD patients (Dost et al, 2020). Preprocessing the data, we subset human AT2^WT and AT2^KRAS cells along with previously uncharacterized mesenchymal cells (COL1A1^POS VIM^POS) (n = 3331 cells) (Figs. 3E and EV4E–G). In agreement with our organoid and in vivo data, early-stage human AT2^KRAS cells have reduced AT2 lineage-defining gene expression. Furthermore, injury/plasticity gene expression signatures, including *CLDN4* and *KRT8* were robustly upregulated in human AT2^KRAS cells (Figs. 3F and EV4G). This data demonstrates that AT2^KRAS cells utilize a normally transient injury/plasticity^HIGH AT2^LOW state for AT2 self-renewal, which promotes LUAD progression and heterogeneity, and is conserved

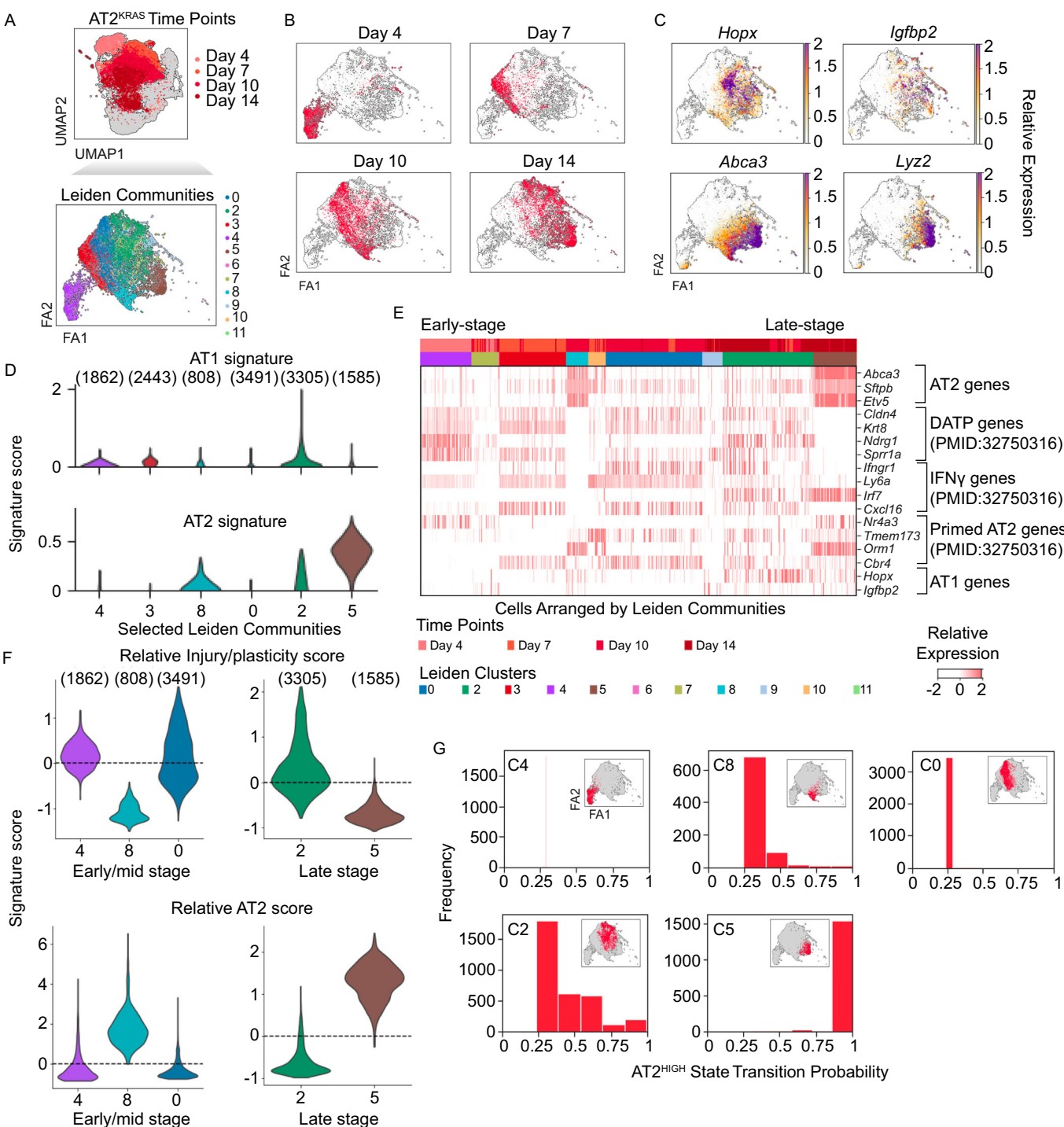

**Figure 2. Kras^G12D promotes distinct temporal subpopulations defined by AT2-lineage and injury/plasticity signatures.**

(**A**) FA2 representation of filtered single cells subset from AT2^KRAS organoid data and their corresponding Leiden community. The FA2 in Fig. 2A is reused in Figs. 2B, C, G and EV1D to show different aspects of the data. (**B**) FA2 representation of filtered single cells subset from AT2^KRAS organoid data and their corresponding time point. (**C**) FA2 representations of filtered single cells subset from AT2^KRAS organoid data and the relative expression of either AT1 or AT2 specific genes. (**D**) AT2 and AT1 signature score level per community using a violin plot (*y*-axis, signature score; *x*-axis, Leiden community). (*n*) denotes the number of cells per cluster. (**E**) Heatmap of relative gene expression in the AT2^KRAS organoid time course data relevant to AT2, AT1, and injury response gene expression signatures. Cells are arranged by time point contributions in each Leiden community; early (left), late (right). (**F**) Relative injury/plasticity and AT2 signature score level per community using a violin plot (*y*-axis, signature score; *x*-axis, Leiden community). (*n*) denotes the number of cells per cluster. (**G**) Transition probabilities of cells in selected AT2^KRAS Leiden communities (*x*-axis; labeled transition probability, *y*-axis; binned frequency of occurrence). Inset highlights the cell states whose probabilities are being represented using a FA2 representation.

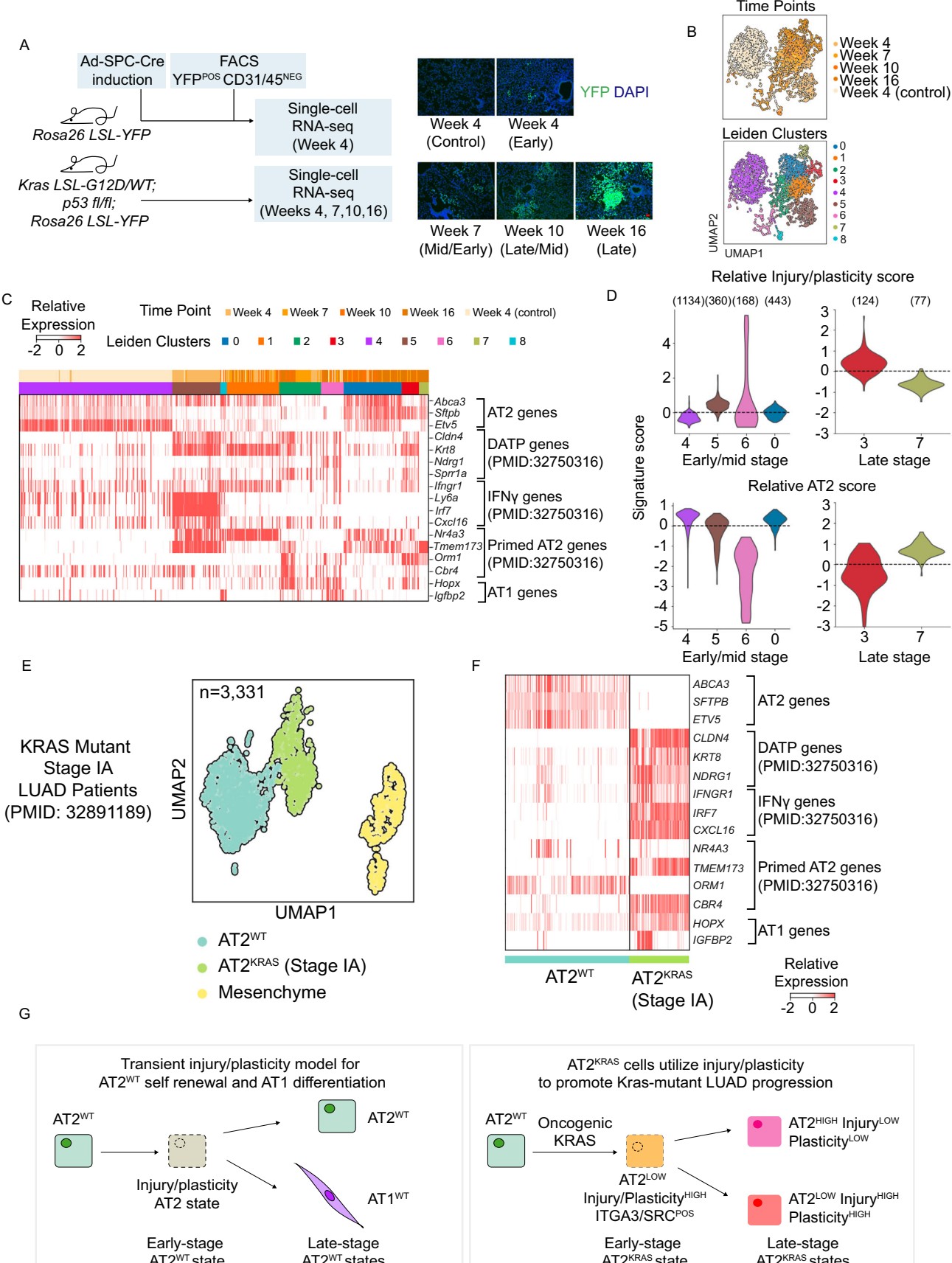

◀

**Figure 3.  AT2^KRAS states are temporally conserved in vivo and in stage IA human LUAD.**

(A) Schematic representation of the in vivo time course experiment. YFP-stained representations from each time point are shown. Scale bar = 100 μm. (B) UMAP representation of filtered single cells from the in vivo experiment, their corresponding time point, and Leiden communities. The UMAP in Fig. 3B is reused in Fig. 4A to show a different aspect of the data. (C) Heatmap of relative gene expression in the in vivo time course data relevant to AT2, AT1, and injury response gene expression signatures. Cells are arranged by time point contributions in each Leiden community; early (left), late (right). (D) Relative injury/plasticity and AT2 signature score level per community using a violin plot (y-axis, signature score; x-axis, Leiden community). (n) denotes the number of cells per cluster. (E) UMAP representation of filtered single cells from previously published stage IA LUAD patients data (Dost et al, 2020) and their corresponding Leiden community (this paper). (F) Heatmap of relative gene expression in human stage IA AT2^WT and AT2^KRAS cells relevant to AT2, AT1, and injury response gene expression signatures. Cells are arranged by Leiden community. (G) Proposed model for AT2^WT and AT2^KRAS transcriptional cell states over time conserved in murine organoids and in vivo. A conserved injury/plasticity state precedes both AT2 self-renewal and AT1 differentiation, which is co-opted by oncogenic KRAS to drive LUAD initiation, heterogeneity, and progression. Importantly, the cell states have distinct transcriptomes in each time point and condition (represented by different colors). Source data are available online for this figure.

in AT2 cells in vivo, in organoids, and human patients with early-stage LUAD (Fig. 3G).

## A KRAS–ITGA3–SRC axis drives the earliest stages of LUAD initiation

Having determined that an injury/plasticity state is shared by self-renewing and tumorigenic AT2 cells, we next sought to determine if *Kras*-driven cells utilize pathways to drive tumorigenesis that are distinct from normal progenitor cell activity and differentiation. We performed DE analysis based on Leiden communities in our AT2 organoid, GEMM, and stage IA human data (Figs. 4A and EV5A and Dataset EV4). DE genes were collected from early-stage states from each dataset: Week 4 GEMM in vivo, Day 4 organoids, and AT2^KRAS cells from stage IA LUAD patients. Lastly, we filtered for statistically significant ($p$ value < 0.05) DE genes identified as receptors (Skelly et al, 2018) (Fig. 4A). Our analysis unbiasedly identified eight receptors differentially expressed by early-stage Kras-mutant cells across all three model systems: *OCLN, ITGA3, ADIPOR1, PLXNB2, CLDN4, ST14, ITGB1,* and *LSR* (Fig. 4B). Importantly, in the GEMM and organoid datasets with temporal resolution, the majority of identified receptors had the highest relative expression in early-stage AT2^KRAS time points, and elevated relative expression in AT2^KRAS cells compared to AT2^WT cells in our stage IA human data (Fig. 4B). In a separate analysis we used Genewalk, a machine-learning (ML) approach to find highly relevant genes and Gene Ontology (GO) terms in early-stage AT2^KRAS cells. The graph-based ML generates a biomedical knowledge graph to identify context-specific interconnections between input genes and biological pathways (Ietswaart et al, 2021). We used DE genes upregulated in the early-stage AT2^KRAS (C8) relative to the AT2^WT (C10) organoid communities as input for ML analysis (Dataset EV5). Genewalk identified integrin subunits *Itga3* and *Itgb1* as receptors with the highest gene-gene and gene-GO interconnections (Fig. 4C) within our identified receptors, prompting us to further investigate the conserved role of integrins in early-stage AT2^KRAS cells. Integrins are heterodimeric receptor complexes that comprise an alpha and beta subunit. There are 24 known human integrin heterodimers that have evolved specialized functions (Ley et al, 2016; Lin et al, 2022; Reed et al, 2015). We chose to focus on the alpha subunit *Itga3* because the role of α3 integrins in early-stage LUAD is poorly understood. To validate the observation that early-stage AT2^KRAS cells have high expression of *Itga3*, RNAScope (Wang et al, 2012), an in situ hybridization technique, was performed using *Itga3* specific probes combined with IF staining for YFP, a marker of AT2^KRAS cells, and

the AT2 marker SPC. Consistent with scRNA-seq data, *Itga3* had the strongest signal in YFP^POS cells at 4 weeks (early) compared to 10 weeks (mid-stage) time point in vivo, with little to no *Itga3* signal in wildtype AT2 cells (YFP^NEG SPC^POS) (Fig. 4D). After confirming that high *Itga3* expression is specific to early-stage AT2^KRAS cells, we examined gene-GO associations identified by our ML analysis. The top GO hit for *Itga3* was "integrin alpha3 beta1 complex" (Fig. 4E). Intriguingly, *Itgb1* was also a conserved receptor hit (Fig. 4B), suggesting a possible function of α3β1 integrin in early-stage AT2^KRAS cells. We observed a positive correlation between the expression of *Itga3* and the injury/plasticity signature at early-stage AT2^KRAS time points, when *Itga3* levels were most significantly DE, and also at mid and later stages (Fig. EV5B).

We next probed what may be downstream of integrin signaling in early AT2^KRAS cells. Further analysis of our ML results revealed proto-oncogene tyrosine kinase SRC as the fifth most interconnected hit in early AT2^KRAS cells (Fig. 4C and Dataset EV5). SRC is a known downstream signal of integrins in leukocytes (Mócsai et al, 2006; Ley et al, 2016), but with uncharacterized function in AT2 cells. The top SRC GO hit identified by our ML analysis was "integrin-mediated signaling pathway" (Fig. 4E). The known connection between SRC and integrin signaling in non-lung cells, the strength of the SRC hit in our ML analysis, and the ML-identified association between SRC and integrin signal made it an ideal candidate for further investigation. We found a positive correlation between *Itga3* and *Src* in AT2 cells at different time points (Fig. EV5C). Next, we calculated the percentage of AT2 cells double positive for *ITGA3* and *SRC* (relative expression >0). *Itga3^POS Src^POS* AT2 cells were enriched in early-stage AT2^KRAS time points across all models examined in this study (Fig. 4F).

Lastly, we investigated if the KRAS–ITGA3–SRC axis in early AT2^KRAS cells could be exploited therapeutically in early-stage LUAD. Small molecule inhibitors of the GDP KRAS-OFF state are emerging, such as sotorasib (LUMAKRAS) and adagrasib (KRAZATI) for KRAS^G12C mutant cancers, and MRTX1133 for KRAS^G12D mutant cancers (Wang et al, 2022). Based on our findings, we reasoned that combining KRAS^G12D and inhibitors of ITGA3 or SRC will improve the efficacy of KRAS inhibitors. Due to the lack of drugs that directly target ITGA3, we inhibited Src using the FDA-approved small molecule dasatinib (SPRYCEL). AT2^KRAS organoids treated at Day 0 with vehicle (DMSO) or adagrasib had little impact on growth because the organoids tested are KRAS^G12D mutant. Some impact on the size of AT2^KRAS organoids was observed using 100 nM of dasatinib (Hochhaus and Kantarjian, 2013) or MRTX113 as a monotherapy. In contrast, a combination of dasatinib with MRTX1133 dramatically suppressed AT2^KRAS

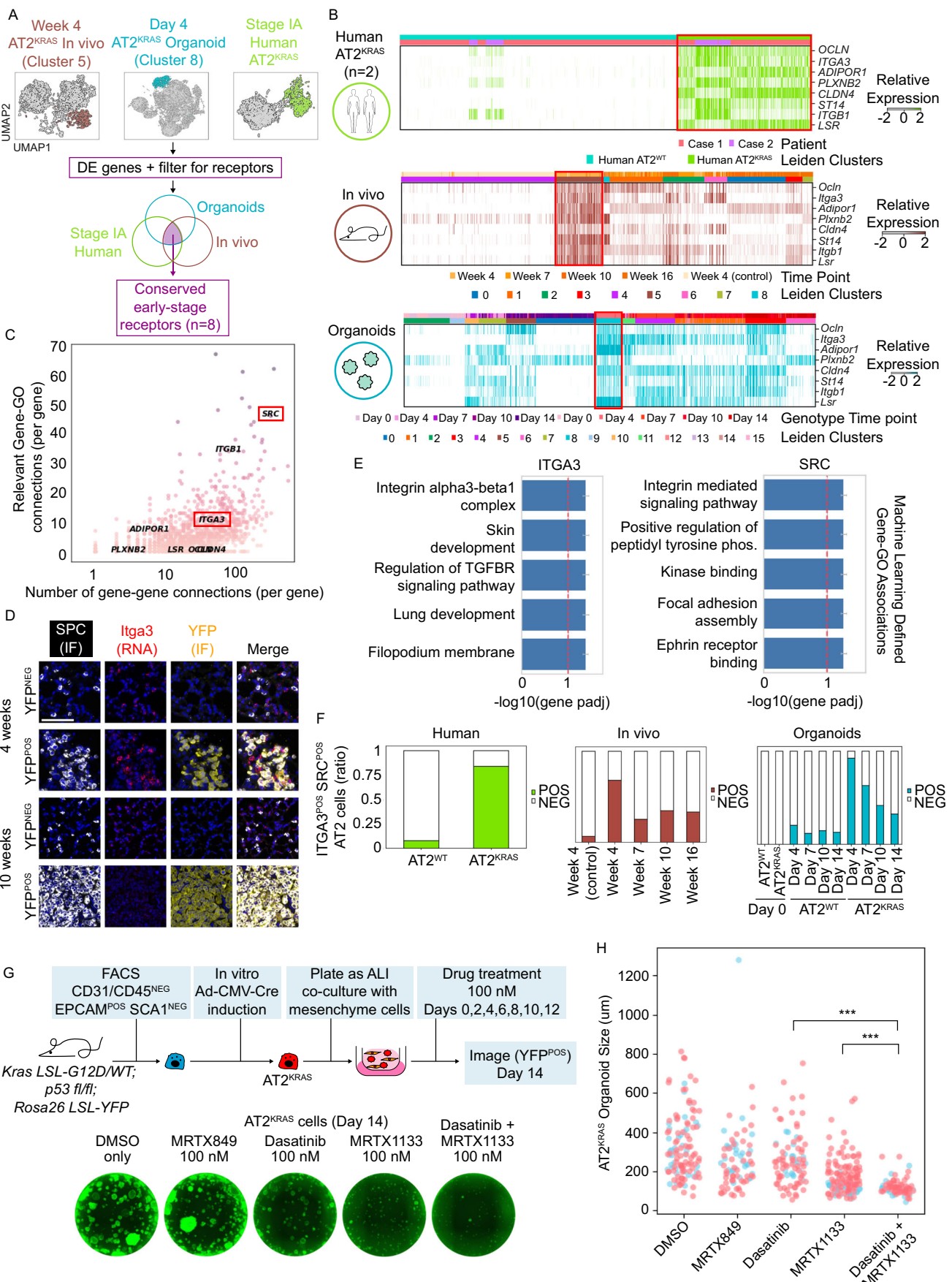

**Figure 4.  A KRAS–ITGA3–SRC axis drives the earliest stages of LUAD initiation.**

(A) Analysis design to identify early-stage AT2^KRAS receptors conserved across species and model system examined in this study. (B) Relative expression of identified receptors in the AT2 organoids, in vivo, and human patient data, using a heatmap (x-axis; Leiden communities arranged by time point contributions, y-axis; conserved receptors). (C) Interconnections between DE genes in early-stage AT2^KRAS organoids identified using machine learning, represented as a dot plot (x-axis; number of gene–gene connections, y-axis; number of gene-GO term connections). Genes selected for further study are highlighted with red boxes. (D) RNAScope analysis in 4-week and 10-week murine lung section following in vivo induction with Ad5SPC-Cre. Scale bar = 100 μm. (E) Bar plot of context-relevant GO terms associated with *Itga3* and *Src* in early-stage AT2^KRAS organoids, identified using machine learning. GO terms are arranged by statistical significance. GeneWalk methodology, error, and statistical methods are described in (Ietswaart et al, 2021). (F) Bar plot representing the percentage of *Itga3*^POS *Src*^POS cells in each Leiden community in the AT2 organoid, in vivo, and human patient datasets. (G) Experimental design of AT2^KRAS organoid co-cultures treated with DMSO (vehicle), MRTX849, MRTX1133, Dasatinib, or MRTX1133/Dasatinib combination, and representative images of D14 AT2^KRAS organoids from each treatment group. (H) Quantitation of Day 14 AT2^KRAS organoid size (diameter, μm) in each treatment group; A total of 80–137 organoids were measured from two independent experiments, represented as blue and red circles in the graph. ***p < 0.001. DMSO vs. all conditions were statistically significant p < 0.05 (not represented on the graph). Statistical significance was determined using a Wilcoxon Rank-Sum Test. Source data are available online for this figure.

organoid growth (Fig. 4G,H). Thus, both KRAS^G12D and SRC signaling are important for early AT2^KRAS growth and could provide a rationale for combination adagrasib/dasatinib treatment as a potential adjuvant or neo-adjuvant treatment for early-stage KRAS^G12D mutant LUAD patients.

## Discussion

These data define a conserved injury-associated, lineage-plastic transitional cell state activated by AT2^WT cells prior to AT2 self-renewal that is co-opted by oncogenic KRAS to drive tumorigenesis. It has long been established that AT2 cells have facultative regenerative capacity and AT1 differentiation potential. A model was proposed for lung development in which AT2 and AT1 cells form independently via a bipotent progenitor cell (Desai et al, 2014; Treutlein et al, 2014). More recently, KRT8^POS DATP/PATS, injury-associated cell states have been observed as intermediate states in the process of AT2-to-AT1 differentiation (Strunz et al, 2020; Choi et al, 2020; Kobayashi et al, 2020). Our data suggest that AT2^WT cells do not directly self-renew, but rather undergo transient lineage suppression prior to AT2 self-renewal. Additionally, aberrant retention of this injury/plasticity state defines KRAS^G12D AT2 cells. However, we do not observe an AT2^HIGH Injury/Plasticity^LOW state in our early-stage LUAD patient data, suggesting there are subtle differences in LUAD progression in humans and mice. Recent studies have highlighted cell-intrinsic lineage plasticity and injury-associated epigenetic rewiring programs in early-stage pancreatic cancer (Burdziak et al, 2023; Alonso-Curbelo et al, 2021). Recently, Han et al also independently identified KRT8^POS alveolar intermediate cells (KACs) with increased plasticity in early-stage KRAS-mutant LUAD (Han et al, 2024), confirming and validating the importance of injury/plastic AT2 intermediates in LUAD initiation.

The KRAS-mutant transitional cell state we describe is enriched for ITGA3^POS SRC^POS cells at the earliest stages of LUAD initiation, and this state can be targeted with clinically relevant small molecules in organoid co-culture systems, implicating "outside-in" integrin and downstream SRC signaling in early LUAD. β3 Integrins have been linked with KRAS to promote fitness and survival of established pancreatic, lung, and breast carcinomas and can promote resistance to EGFR inhibitors (Seguin et al, 2014). Effective inhibitors of integrin signaling remain limited (Lin et al, 2022); therefore, other components of integrin signaling pathways may provide novel therapeutic opportunities. The use of SRC

inhibitors like dasatinib as a monotherapy for late-stage NSCLC failed Phase II clinical trials (Johnson et al, 2010). Our work provides a rationale for re-testing SRC inhibitors focused on early-stage LUAD in combination with KRAS^G12D inhibitors such as MRTX1133. Notably, we cannot rule out the possibility that SRC inhibition might enhance the ability of MRTX1133 to bind to mutant KRAS. Recent findings showed that inhibiting KRAS induced an AT1-like state in wildtype AT2 cells and LUAD, suggesting that alveolar differentiation is a resistance mechanism for therapies targeting the RTK–KRAS axis (Li et al, 2023). We are currently investigating whether the mechanism of the effect of the dual RTK-KRAS inhibition on KP organoid growth is via inducing an AT1 state or reduction of the injury/plasticity state. Our results demonstrate that dual RTK-KRAS inhibition potently inhibits early-stage LUAD. Beyond currently available strategies, our study suggests that targeting the unique injury-associated transitional cell states driven by oncogenic mutations can provide a new means to intervene in early-stage LUAD.

## Methods

### Mouse cohorts

*Kras*^*LSL-G12D/WT*; *p53*^*flox/flox* (Jackson et al, 2005) mice were crossed to mice homozygous for the *Rosa26*^*LSL-YFP* allele to obtain *Kras*^*LSL-G12D/WT*; *p53*^*flox/flox*; *Rosa26*^*LSL-YFP* mice. Mice were maintained in virus-free conditions. All mouse experiments were approved by the BCH Animal Care and Use Committee, accredited by AAALAC, and were performed in accordance with relevant institutional and national guidelines and regulations.

### Stage IA LUAD patient information

As described previously (Dost et al, 2020), samples from two patients with a stage IA LUAD diagnosis were analyzed in this study. One patient was female, 74 years old, with a KRAS G12F mutation. The other patient was female, 77 years old, with a KRAS G12V mutation. All patients provided written informed consent. The studies were approved by the UCLA institutional review board.

### In vivo adenovirus infection

Young adult mice (less than 6 months) were infected with $6 \times 10^8$ PFU adenovirus containing the AT2-specific SPC promoter by

intratracheal instillation as described previously (DuPage et al, 2009). In this study, only female mice were used. To avoid technical and batch variation when generating scRNA-seq libraries, the mice were induced at different times, allowing cells to be collected via FACS at the same time. The 16-week time point mice were induced first, followed by inducing the 10-week group six weeks later, the 7-week group 3 weeks later, and finally the 4-week group 3 weeks later.

## Generating a single-cell suspension from murine lungs

Mice were anesthetized with avertin, lungs were perfused intratracheally with 10 mL PBS, followed by intratracheal instillation of 2 mL dispase (Corning). Lungs were placed on iced, minced, and incubated in 0.0025% DNase (Sigma Aldrich) and 100 mg/mL collagenase/dispase (Roche) in PBS for 45 min at 37 °C, filtered through 100 mm and 40 mm cell strainers (Fisher Scientific), and centrifuged at 1000 rpm, 5 min at 4 °C. Cells were resuspended in red blood cell lysis buffer (0.15 M NH4Cl, 10 mM KHCO3, 0.1 mM EDTA) for 1.5 min, washed with advanced DMEM (GIBCO), and resuspended in PBS/10% FBS (PF10) at 1 million cells/100 μL.

## Isolation of primary AT2 cells from murine lungs using FACS

Lung single-cell suspensions, see "Generating a single-cell suspension from murine lungs" were incubated for 10 min on ice with DAPI as a viability dye and the following antibodies: anti-CD31 APC, anti-CD45 APC, anti-Ly-6A/E (SCA1) APC/Cy7 (all Fisher Scientific), and anti-CD326 (EPCAM) PE/Cy7 (Biolegend) (all 1:100). Single stain controls and fluorophore minus one (FMO) controls were included for each experiment. FACS was performed using a FACSAria II to collect CD31$^{NEG}$ CD45$^{NEG}$ EPCAM$^{POS}$ SCA1$^{NEG}$ cells. Cells were collected in PBS/10% FBS, and analysis was performed with FlowJo.

## In vitro infection of primary lung epithelial cells using adenovirus for organoid co-culture

Murine AT2 cells (CD31$^{NEG}$ CD45$^{NEG}$ EPCAM$^{POS}$ SCA1$^{NEG}$) cells isolated by FACS as described in "Isolation of primary AT2 cells from murine lungs using FACS" were pelleted by pulse spin, the supernatant removed, then resuspended in 100 μL per 100,000 cells of MTEC/Plus media (Zhang et al, 2017) containing 6× 10$^7$ PFU/mL of Ad5CMV-Cre. The cells were incubated for 1 h at 37 °C, 5% CO$_2$ in 1.5 mL tubes. Cells were then pelleted by pulse spin and resuspended in 1× phosphate-buffered saline (PBS). This step was repeated twice for a total of three washing steps. Cells were resuspended in DMEM/F12 (Invitrogen) supplemented with 10% FBS, penicillin/streptomycin, 1 mM HEPES, and insulin/transferrin/selenium (Corning) (3D media) at a concentration of 5,000 live cells (trypan blue negative) per 50 μL. For supporting cells, a mix of neonatal stromal/mesenchyme cells was isolated as described previously (Lee et al, 2014). The supporting cells were pelleted and resuspended in growth factor reduced (GFR) Matrigel at a concentration of 50,000 cells per 50 μL. Equal volumes of cells in 3D media and supporting mesenchyme cells in GFR Matrigel were combined and 100 μL were pipetted into a Transwell (Corning). Plates were incubated for 20 min at 37 °C, 5% CO$_2$ until Matrigel

solidified. Finally, 500 μL of 3D media was added to the bottom of the transwell, and 3D media was changed every 2–3 days.

## AT2$^{KRAS}$ organoid drug treatment

In vitro induced organoids were plated as described in "In vitro infection of primary lung epithelial cells using adenovirus for organoid co-culture." Starting Day 0, organoids were grown in 3D media supplemented with 100 nM of dasatinib (SRC inhibitor), MRTX1133 (KRAS G12D inhibitor), MRTX849 (KRAS G12C inhibitor), a combination of dasatinib and MRTX1133, or DMSO only (vehicle). 500 μL of media was added to the base of the transwell. Drug-treated 3D media was changed every two days. Fresh stocks of drug-treated 3D media were made every 4 days.

## Collecting cells for scRNA-seq using FACS

For in vivo experiments, cells were collected from mice infected with Ad5SPC-Cre at the same time, see "In vivo adenovirus infection". Single-cell suspensions were generated, see "Generating a single-cell suspension from murine lungs" then incubated for 15 min on ice and protected from light exposure, with 4′,6-diamidino-2-phenylindole (DAPI) as a viability dye and the following antibodies: anti-CD31 PE (BD Biosciences), anti-CD45 APC (Fisher Scientific), anti-CD326 (EPCAM) PE-Cy7 (Biolegend) (all 1:100). Single stain controls and fluorophore minus one (FMO) controls were included for each experiment. FACS was performed on a FACSAria II, cells collected in PBS/10% FBS, and analysis was done with FlowJo.

For organoid experiments, single cell suspensions were obtained at days 4, 7, 10, and 14 for FACS, cells were incubated with EPCAM-PeCy7 (BioLegend) and DAPI (Sigma-Aldrich) for 10 min on ice. A DAPI only control served as the fluorophore minus one (FMO) control for EPCAM. FACS was performed on a FACSAria II and analysis was done with FlowJo.

## Organoid and in vivo scRNA-seq

ScRNA-Seq was performed using the 10× Genomics platform (10× Genomics, Pleasanton, CA). FACS sorted cells from either mice or organoid cultures were encapsulated with a 10× Genomics Chromium Controller Instrument using the Chromium Single Cell A Chip Kit. Encapsulation, reverse transcription, cDNA amplification, and library preparation reagents are from the Chromium Single Cell Library & Gel Bead Kit v3. Briefly, single cells were resuspended in PF10 at a concentration of 1000 cells/μL. The protocol was performed as per 10× Genomics protocols without modification (chromium single cell 3′ reagent kits user guide v3 chemistry). Total cDNA and cDNA quality following amplification and clean-up was determined using a QubitTM dsDNA HS assay kit and the Agilent TapeStation High Sensitivity D5000 ScreenTape System. Library quality pre-sequencing was determined using Agilent TapeStation and QPCR. TapeStation analysis and library QPCR was performed by the Biopolymers Facility at Harvard Medical School. Libraries were sequenced using an Illumina NextSeq500 using paired-end sequencing with single indexing (Read 1 = 28 cycles, Index (i7) = 8 cycles, and Read 2 = 91 cycles). Reads were aligned to the mm10-1.2.0 reference genome and count matrices were generated using CellRanger 3.1.0 (10× Genomics).

## Single-cell RNA-seq data analysis

Jupyter notebooks containing all computational analysis in this study can be found on GitHub (https://github.com/alm8517/KRASmutant_TimeSeries). In brief, count matrices generated by CellRanger were read into the Python single-cell analysis environment Scanpy (Wolf et al, 2018). For the organoid co-culture and human data, cells with >20% mitochondrial content, which correlated with low read count were removed. The data was normalized, logarithmized, and the significant number of principle components determined using in-built Scanpy functions. Data was denoised using Markov Affinity-based, Graph Imputation (MAGIC) with the following parameters (Gene to return = all, $k = 5$, $t = 5$, n_pca = 30) (van Dijk et al, 2018) followed by nearest neighbor calculation and UMAP dimensionality reduction. Organoid data was subset into epithelial cells and mesenchymal cells for further analysis. For in vivo data, cells with >10% mitochondrial content, which correlated with low read count, were removed. The data was normalized, logarithmized, and the significant number of principle components was determined using in-built Scanpy functions. Next, we performed nearest neighbor calculations and UMAP dimensionality reduction. Cell types were annotated and epithelial cells were subset for further analysis. A reference list of murine transcription factors was curated from two publicly available datasets; The Animal Transcription Factor Database (Hu et al, 2019) and the TcoF-DB v2 (Schmeier et al, 2017). Receptor-ligand relationships were curated from the CellPhoneDB database (Efremova et al, 2020) and a previously published curated list of receptor-ligand interactions (Skelly et al, 2018). AT1 and AT2 marker genes are from the Panglao database (Franzén et al, 2019). Relative (early/mid and late) injury/plasticity (*Cldn4, Krt8, Nr4a3, Ifngr1, Ndrg1, Sox9, Hmga2, Itga2*) and AT2 signatures (*Etv5, Lyz2, Abca3*) were calculated using a subset of signature marker genes defined in the figures and associated Jupyter notebooks. Mesenchyme population markers are from various published papers (Lee et al, 2017; Zepp et al, 2017; Xie et al, 2018). Machine learning analysis was performed as previously described, using Genewalk v1.5.3 (Ietswaart et al, 2021).

## Staining and IF of organoid cultures and in vivo lung sections

To image whole wells, multiple overlapping images of live organoid cultures were taken at various time points and stitched together using EvosTM FL Auto2 software. Images were processed using FIJI (Schindelin et al, 2012). To image YFP-positive cells in murine lung section, a lobe of lung was clamped and removed during the preparation of the Ad5SPC-Cre treated lungs for scRNA-seq. YFP staining of slides was performed as previously described (Dost et al, 2020).

## RNAScope of in vivo induced murine lung sections

Sectioned lung tissues underwent deparaffinization by incubation with xylene and rehydration in 100% ethanol. The deparaffinized slides were blocked in RNAscope hydrogen peroxide for 10 min at room temperature. Fresh 1× Target Retrieval Reagents were prepared by adding 630 mL of distilled water to 70 mL of 10× Target Retrieval Reagents in a beaker. The Target Retrieval buffer

was heated to a mild boil (98–102 °C) and maintained at a temperature within the range of 98–102 °C for Target Retrieval. The slides were submerged in the heated 1× RNAscope Target Retrieval solution and boiled for 15 min. After Target Retrieval, the slides were washed in distilled water and 100% ethanol before drying at room temperature for 5 min. Hydrophobic barriers were created around each section using the ImmEdge hydrophobic barrier pen which were allowed to dry for 5 min. The slides were treated with RNAscope Protease Plus and incubated at 40 °C for 30 min in the HybEZ Oven. Meanwhile, the 120 mL of RNAscope 50X Wash buffer was warmed to 40 °C for 10–20 min before being mixed with 5.88 L distilled water in a large carboy to prepare 6 L of 1× Wash Buffer. The ready-to-use 1× C1 probe (for labeling ITGA3) was warmed for 10 min at 40 °C then cooled to room temperature before use. After the RNAscope Protease Plus incubation period, the protease plus was removed from each section and the C1 probe was immediately applied to entirely cover each section. The C1 probe hybridized for 2 h at 40 °C in the HybEZ Oven. After probe hybridization, the slides were washed in 1× Wash Buffer and stored overnight in 5X Saline Sodium Citrate (SSC) at room temperature. Reagents AMP1, AMP2, AMP3, HRP-C1, and HRP-blockers (RNAscope Multiplex Fluorescent Reagent Kit v2) were equilibrated at room temperature before use. Excess liquid was removed from the slides and each slide received enough drops of RNAscope Multiplex FL v2 Amp 1 to entirely cover the sections. Slides were incubated in the HybEZ Oven for 30 min at 40 °C then washed in 1× Wash Buffer before the addition of RNAscope Multiplex FL v2 Amp 2 to each section. The slides were once more incubated in the HybEZ Oven for 30 min at 40 °C. After a brief rinse in 1× Wash buffer, the slides received enough drops of RNAscope Multiplex FL v2 Amp 3 to entirely cover each section before being incubated in the HybEZ Oven for 15 min at 40 °C. Meanwhile, the TSA Vivid 570 fluorophore for labeling the C1 probe was prepared to a dilution of 1:1500 in TSA buffer. To develop the HRP-C1 signal, drops of RNAscope Multiplex FL v2 HRP-C1 were added to entirely cover each slide and incubated in the HybEZ Oven for 15 min at 40 °C. After a brief rinse in 1× Wash Buffer, each section received approximately 150–200 μL diluted fluorophore and the slides incubated for 30 min at 40 °C. A brief rinse of the slides in 1× Wash Buffer followed incubation and RNAscope Multiplex FL v2 HRP blocker was applied to each section. The slides were incubated for 15 min at 40 °C. Slides were washed in PBS/0.2% Triton X-(PBS-T) and blocked with 10% normal donkey serum for 1 h at room temperature. Primary antibodies were incubated overnight at 4°C at the indicated dilutions: goat anti-YFP (1:400, Abcam) and rabbit anti-SPC (1:1000, Abcam) in PBS/0.2% Triton X-(PBS-T). After 3× washing, slides were incubated with Alexa Fluor-coupled secondary antibodies for 1 h at room temperature - goat 488 and rabbit 647 (all Invitrogen, 1:200). After 3× washing, slides were mounted using Prolong Gold with DAPI (Invitrogen). Images used for RNAscope/IF analysis were captured at consistent exposure times, which were determined with negative controls for primary antibodies.

## Data availability

All raw and processed scRNA-seq data were deposited and made publicly available on the NCBI Gene Expression Omnibus (GEO)

and the Sequencing Read Archive under GEO accession: GSE253461 (https://www.ncbi.nlm.nih.gov/geo/query/acc.cgi?acc=GSE253461). Jupyter notebooks containing all bioinformatics analysis are available on GitHub: https://github.com/alm8517/Kras_timecourse_study.

The source data of this paper are collected in the following database record: biostudies:S-SCDT-10_1038-S44318-024-00113-5.

## Peer review information

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

## Acknowledgements

We thank members of the Kim Lab for helpful discussion and feedback. We thank Dr. Ronald Mathieu, Mahnaz Paktinat, and the flow cytometry core facility at Boston Children's Hospital (BCH), the single-cell core facility at Harvard Medical School (HMS), the Zon lab for the use of their 10× Genomics Chromium Controller, the HMS biopolymers facility, the DFHCC rodent histopathology facility, and the BCH Pathology Department histology lab. This work was supported by the Damon Runyon Cancer Research Foundation Postdoctoral Fellowship (no. DRG:2368-19) and a Burroughs Wellcome Fund Postdoctoral Enrichment Program Award (no. 1019903) (ALM), and in part by the Hope Funds for Cancer Research Postdoctoral Fellowship (SML), a Boehringer Ingelheim Fonds PhD fellowship (AFMD), a Damon Runyon Cancer Research Foundation Postdoctoral Fellowship (no. 2523-24) (SG) and R01 HL090136, R01 HL132266, R01 HL125821, U01 HL100402, RFA-HL-09-004, R35HL150876, LONGFONDS Accelerate, project BREATH, R01CA216188, P50CA265826 (SPORE), R01CA233671 American Cancer Society Research Scholar Grant RSG-08-082-01-MGO, the V Foundation for Cancer Research, the Thoracic Foundation, the Ellison Foundation, the American Lung Association LCD-619492 R35HL150876, The G. Harold & Leila Y. Mathers Foundation, and the Harvard Stem Cell Institute (CFK).

## Author contributions

**Aaron L Moye**: Conceptualization; Resources; Data curation; Software; Formal analysis; Supervision; Funding acquisition; Validation; Investigation; Visualization; Methodology; Writing—original draft; Project administration; Writing—review and editing. **Antonella FM Dost**: Conceptualization; Data curation; Formal analysis; Investigation; Methodology. **Robert Ietswaart**: Software; Formal analysis. **Shreoshi Sengupta**: Investigation. **VanNashlee Ya**: Data curation; Formal analysis. **Chrystal Aluya**: Data curation. **Caroline G Fahey**: Data curation; Formal analysis. **Sharon M Louie**: Data curation; Formal analysis. **Margherita Paschini**: Data curation; Formal analysis. **Carla F Kim**: Conceptualization; Resources; Data curation; Formal analysis; Supervision; Funding acquisition; Investigation; Methodology; Writing—original draft; Project administration; Writing—review and editing.

Source data underlying figure panels in this paper may have individual authorship assigned. Where available, figure panel/source data authorship is listed in the following database record: biostudies:S-SCDT-10_1038-S44318-024-00113-5.

## Disclosure and competing interests statement

ALM, RI, and CFK are founders of Cellforma, Inc. CFK is a member of the advisory Editorial Board of *The EMBO Journal*. This has no bearing on the editorial consideration of this article for publication.

# Expanded View Figures

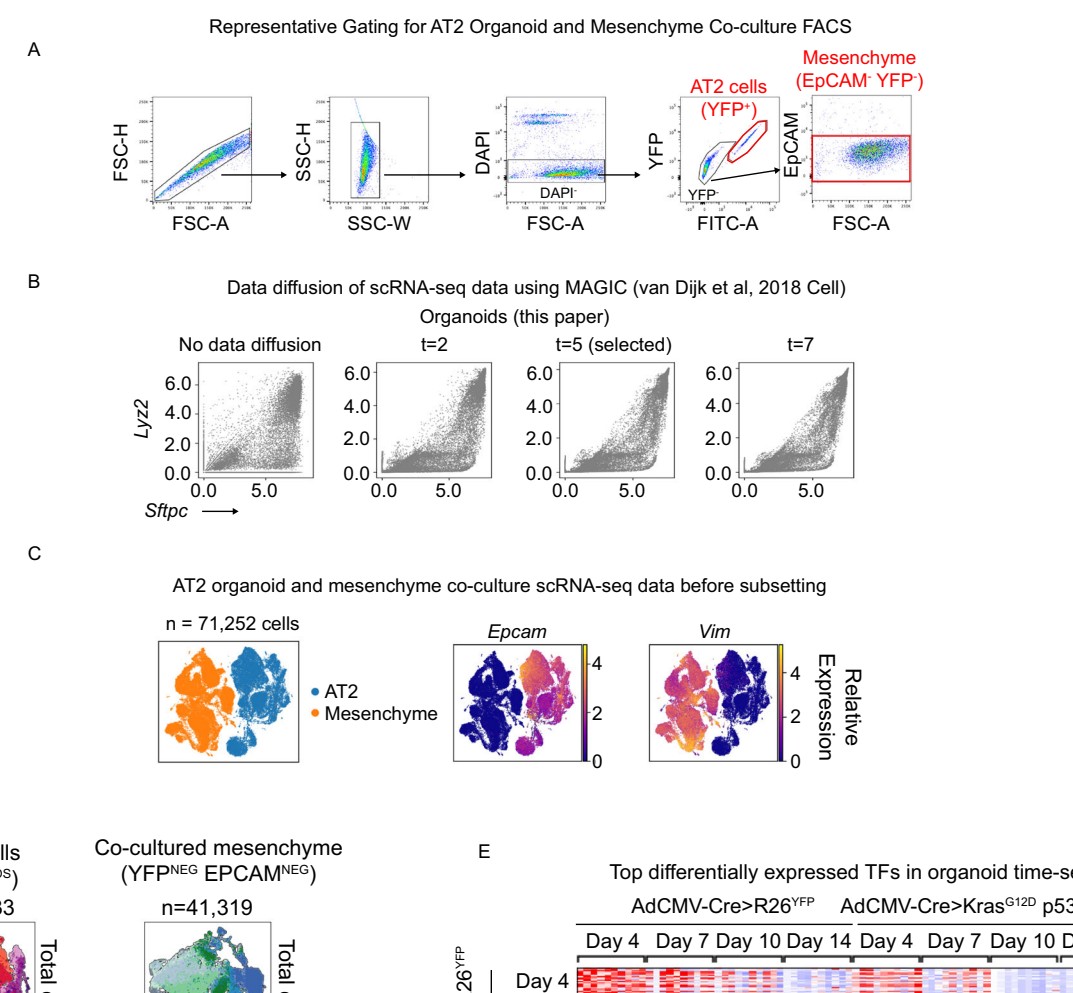

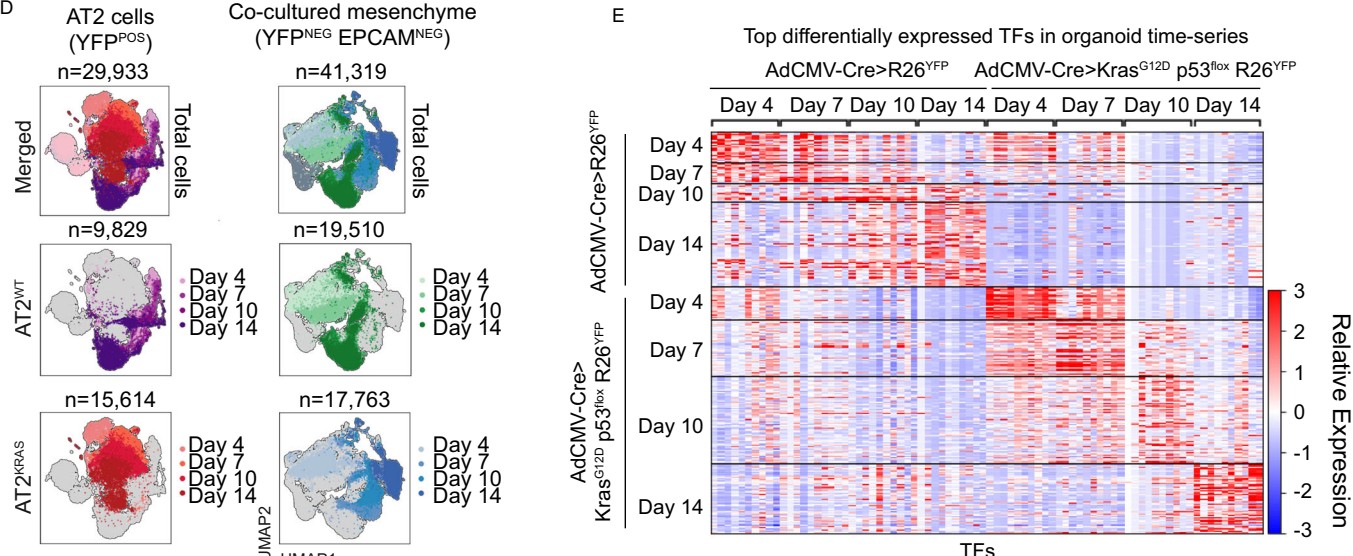

**Figure EV1.  Generation and initial characterization of the AT2-mesenchyme organoid co-culture scRNA-seq dataset.**

(A) Representative FACS plot for AT2-mesenchyme co-culture time course for scRNA-seq analysis. (B) Correlation between *Sftpc* and *Lyz2* expression in individual organoid co-culture cells, after different levels of data diffusion (*t*) (van Dijk et al, 2018). (C) UMAP representation of filtered single cells from organoid co-cultures before subsetting, their corresponding population of origin, and *Epcam* or *Vim* expression. (D) UMAP representations of filtered single cells from organoid co-cultures and their corresponding population of origin, genotype, and time point. (E) Heatmap of the top 10 differentially expressed transcription factors in AT2 cells based on genotype and time point.

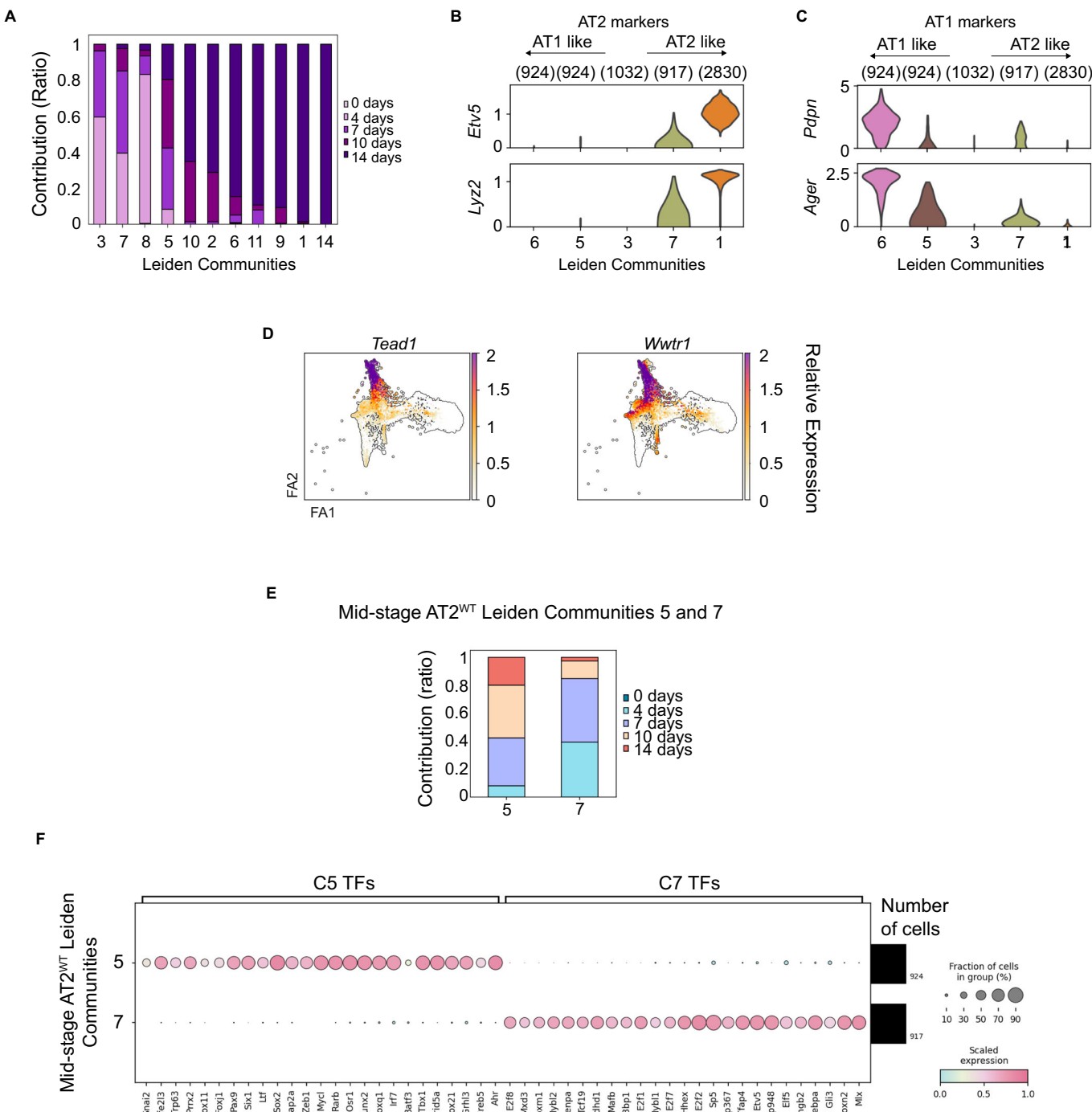

**Figure EV2.  Additional characterization of the AT2^WT organoid scRNA-seq dataset.**

(**A**) Bar plot representing time point contributions in each Leiden community in the AT2^WT organoid scRNA-seq data, represented as a ratio. Leiden communities consisting primarily of Day 0 AT2 cells were excluded from the barplot. (**B**) Relative expression of AT2 genes *Etv5* and *Lyz2* per community using a violin plot (*y*-axis, Leiden community; *x*-axis, relative expression). (*n*) denotes the number of cells per cluster. (**C**) Relative expression of AT1 genes *Pdpn* and *Ager* per community using a violin plot (*y*-axis, Leiden community; *x*-axis, relative expression). (**D**) FA2 representations of filtered single cells subset from AT2^WT organoid data and the relative expression of either *Tead1* or *Wwtr1*. (**E**) Bar plot representing time point contributions to AT2^WT organoid intermediates C5 (AT1 fate) and C7 (AT2 fate). (**F**) Relative expression of the top 25 DE TFs in C5 and C7 AT2^WT intermediate states using a dot plot (*x*-axis; DE genes, *y*-axis; Leiden community).

A

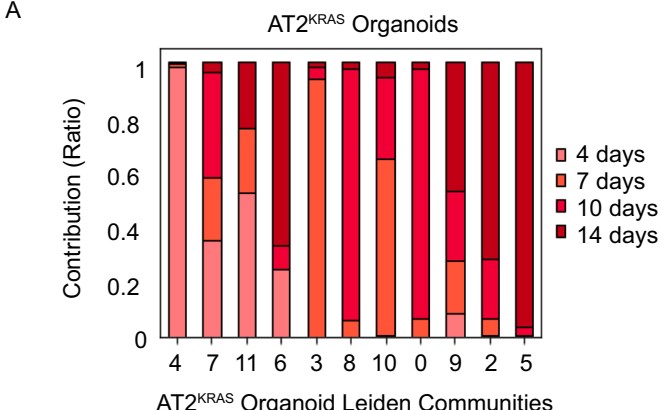

B

**Figure EV3.    Additional characterization of the AT2^KRAS organoid scRNA-seq dataset.**

(**A**) Barplot representing time point contributions in each Leiden community in the AT2^KRAS organoid scRNA-seq data, represented as a ratio. (**B**) Relative *Sox9* expression in select early-, mid-, and late-stage communities using a violin plot (*y*-axis, relative *Sox9* expression; *x*-axis, Leiden community). (*n*) denotes the number of cells per cluster.

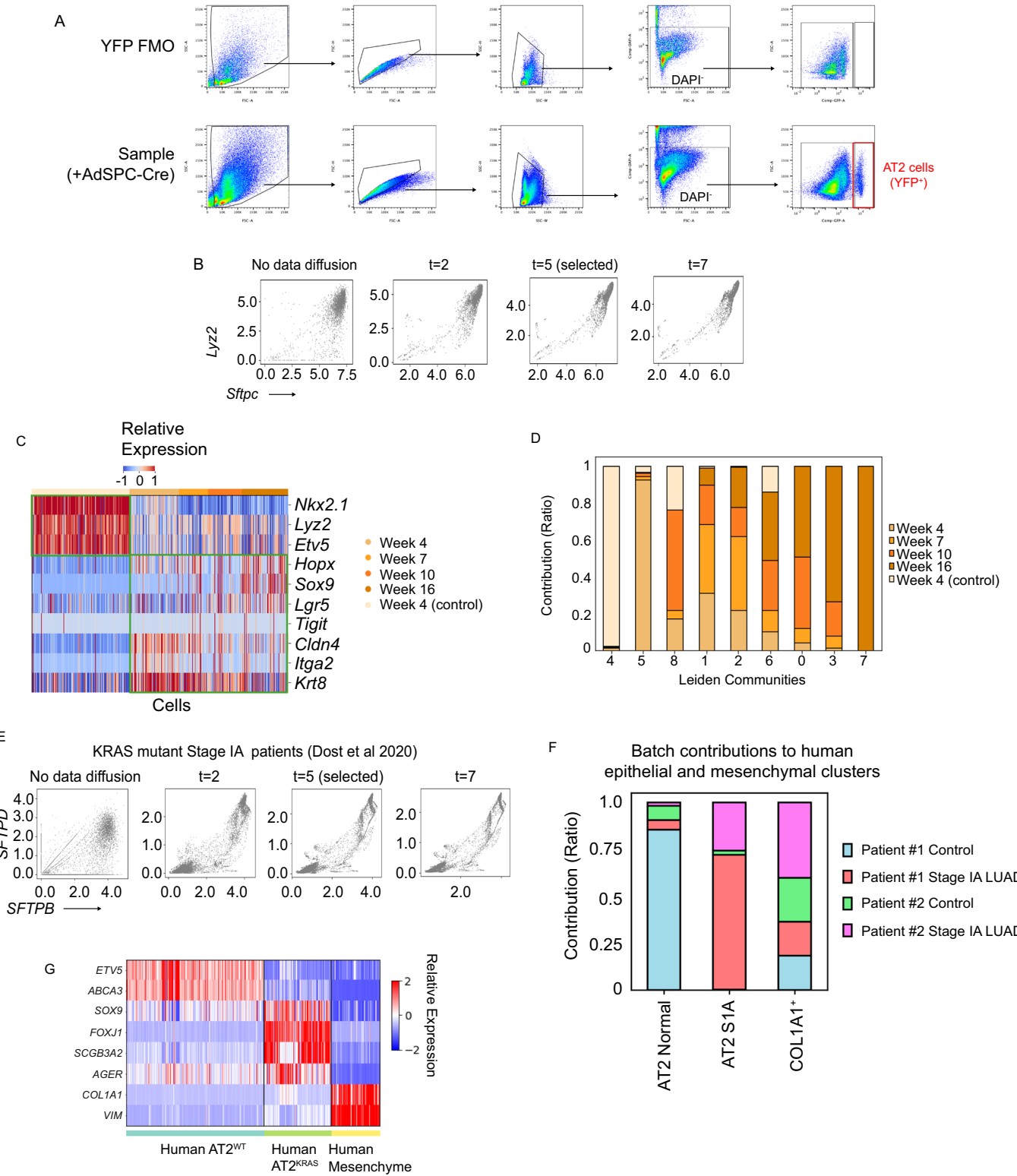

**Figure EV4. Additional characterization of the in vivo and stage IA human scRNA-seq datasets.**

(A) Representative FACS plot for in vivo time course for scRNA-seq experiment. (B) Correlation between *Sftpc* and *Lyz2* expression in vivo, in individual AT2 cells after different levels of data diffusion (*t*) (van Dijk et al, 2018). (C) Heatmap of gene expression in the in vivo time course data relevant to AT2, AT1, development, stem cell, and injury response gene expression signatures. (D) Bar plot representing time point contributions in each Leiden community in the in vivo scRNA-seq data, represented as a ratio. (E) Correlation between *SFTPD* and *SFTPB* expression in individual stage IA human cells after different levels of data diffusion (*t*) (van Dijk et al, 2018). (F) Bar plot representing patient and sample type batch contributions for each Leiden community in the human scRNA-seq data, represented as a ratio. (G) Heatmap of gene expression in human data relevant to different lung lineages and development.

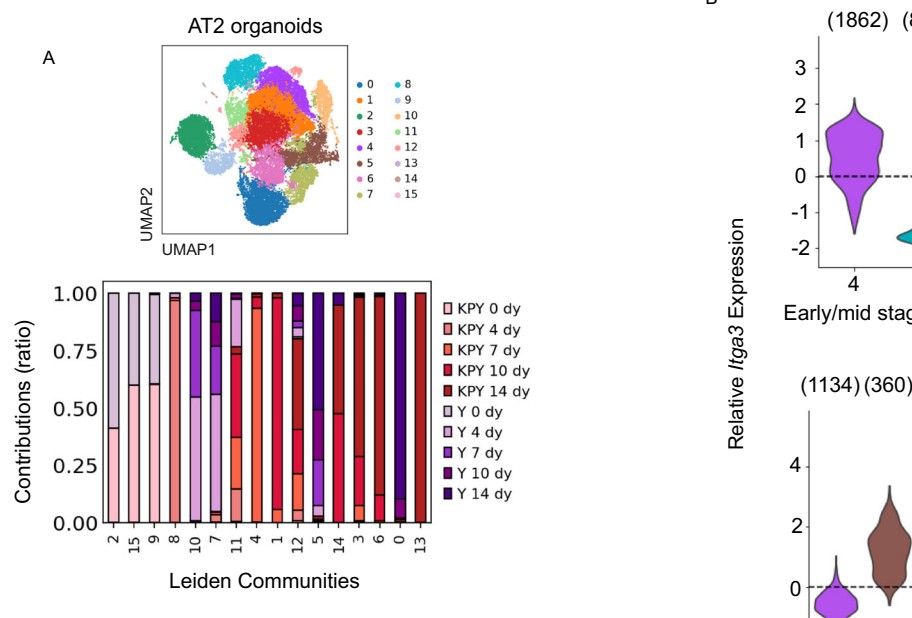

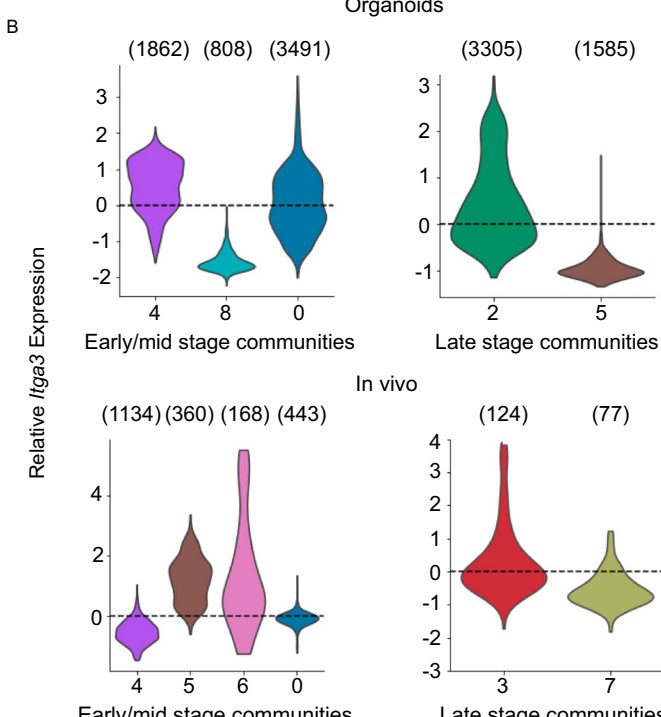

**Figure EV5.   Additional analysis related to Fig. 4.**

(A) UMAP representation of Leiden communities in the combined AT2 organoid data. Bar plot representing time point contributions for each Leiden community, represented as a ratio. (B) Relative *Itga3* expression in AT2$^{KRAS}$ organoids (top) and in vivo (bottom) Leiden communities using a violin plot, subset into early-/mid- and late-stage time points (y-axis, relative *Itga3* expression; x-axis, Leiden community). (C) Correlation between *Src* and *Itga3* relative expression at various time points in AT2 organoids, represented as a scatterplot. Each point represents a single cell.

