## [Peer Review File · The EMBO Journal]

Early-stage lung cancer is driven by a transitional cell state dependent on a KRAS-ITGA3-SRC axis

Aaron Moye, Antonella Dost, Robert Ietswaart, Shreoshi Sengupta, VanNashlee Ya, Chrystal Aluya, Caroline Fahey, Sharon Louie, Margherita Paschini, and Carla Kim

Corresponding author: Carla Kim (carla.kim@childrens.harvard.edu)

Review Timeline:

Submission Date:	29th Feb 24
Editorial Decision:	1st Mar 24
Revision Received:	22nd Mar 24
Editorial Decision:	28th Mar 24
Revision Received:	4th Apr 24
Accepted:	17th Apr 24

Editor: Kelly Anderson

Transaction Report:

(Note: Please note that the manuscript was previously reviewed at another journal and the reports were taken into account in the decision making process at The EMBO Journal. Since the original reviews are not subject to EMBO Press' transparent review process policy, the reports and author response cannot be published. With the exception of the correction of typographical or spelling errors that could be a source of ambiguity, letters and reports are not edited. Depending on transfer agreements, referee reports obtained elsewhere may or may not be included in this compilation. Referee reports are anonymous unless the Referee chooses to sign their reports.)

Dear Dr. Kim,

Thank you for submitting your manuscript for consideration by the EMBO Journal.

As discussed via zoom, please add the final experimental data and update the references and discussion to include the recently published literature in your revised version.

Thank you for the opportunity to consider your work for publication. I look forward to your revision.

Yours sincerely,

Kelly M Anderson, PhD
Editor, The EMBO Journal
k.anderson@embojournal.org

The authors addressed the remaining issues.

Dear Carla,

Congratulations on a great revision! Please attend to the following editorial items and add this as a point by point so that we can move forward with your manuscript:

1. Please include a title page, the abstract and introduction should begin on the next page.
2. Please add up to five keywords, which may or may not appear in the title, should be given in alphabetical order, below the abstract, each separated by a slash (/).
3. Please update the reference format: should be in alphabetical order and et al is needed after 10 author names.
4. Please rename the conflict of interest statement to: "Disclosure Statement and Competing Interest". Please also include the following sentence: Carla Kim is a member of the advisory Editorial Board of The EMBO Journal. This has no bearing on the editorial consideration of this article for publication.
5. Please provide the figures separately as production quality files rather than one PDF.
6. Please rename tables 1-5 to Datasets EvV1-EV5 and upload as datasets. Please also update the callouts to these datasets in the main manuscript accordingly.
7. We include a synopsis of the paper (see <http://emboj.embopress.org/>). Please provide me with a general summary statement and 3-5 bullet points that capture the key findings of the paper.
8. We also need a summary figure for the synopsis. The size should be 550 wide by 200-440 high (pixels). You can also use something from the figures if that is easier.
9. Please ensure the manuscript sections are in the following order: Title page - Abstract & Keywords - Introduction - Results - Discussion - Methods - Data Availability - Acknowledgments - Disclosure Statement & Competing Interests - References - Figure Legends - Tables with legends - Expanded View Figure Legends.
10. Please rename supplemental figures to Figure EV1-EV5 and update the figure legends and callouts in the main manuscript.
11. As discussed via email, please update the figure legends as needed to clarify when any images are used in multiple panels.
12. Please provide the specific URL and reviewer access code for GSE253461 dataset in the Data Availability statement
13. Please provide the accession ID for the NCBI sequencing read archive database in the DAS
14. Please indicate the statistical test used for data analysis in the legends of Fig4e and 4h.
15. In Figure 4h there is a mismatch between the annotated p values in the figure legend and the annotated p values in the figure file.
16. Please add N information to figures 1f, 2d, 2f, 3d, 4e, S2b, S2c, S3b, S5b.
17. Please define the error bars in the legend of figure 4e.
18. Please add a scale bar and its definition to figure 3a.

Thank you for the opportunity to consider your work for publication. I look forward to your revision.

Yours sincerely,

Kelly M Anderson, PhD
Editor, The EMBO Journal
k.anderson@embojournal.org

Further information is available in our Guide For Authors: <https://www.embopress.org/page/journal/14602075/>

authorguide

The authors addressed the minor editorial issues.

Dear Carla,

Congratulations on an excellent manuscript, I am pleased to inform you that your manuscript has been accepted in the EMBO Journal. Thank you for your comprehensive response to the referee concerns and for providing detailed source data. It has been a pleasure to work with you to get this to the acceptance stage.

I will begin the final checks on your manuscript before submitting to the publisher. Once at the publisher, it will take about 3 weeks for your manuscript to be published online (although I will ask them to expedite). As a reminder, the entire review process, including referee concerns and your point-by-point response, will be available to readers.

I will be in touch throughout the final editorial process until publication. In the meantime, I hope you find time to celebrate!

Warm wishes,
Kelly

Kelly M Anderson, PhD
Editor, The EMBO Journal
k.anderson@embojournal.org
